# Innovative Photonic Sensors for Safety and Security, Part I: Fundamentals, Infrastructural and Ground Transportations

**DOI:** 10.3390/s23052558

**Published:** 2023-02-25

**Authors:** Aldo Minardo, Romeo Bernini, Gaia Maria Berruti, Giovanni Breglio, Francesco Antonio Bruno, Salvatore Buontempo, Stefania Campopiano, Ester Catalano, Marco Consales, Agnese Coscetta, Andrea Cusano, Maria Alessandra Cutolo, Pasquale Di Palma, Flavio Esposito, Francesco Fienga, Michele Giordano, Antonio Iele, Agostino Iadicicco, Andrea Irace, Mohammed Janneh, Armando Laudati, Marco Leone, Luca Maresca, Vincenzo Romano Marrazzo, Marco Pisco, Giuseppe Quero, Michele Riccio, Anubhav Srivastava, Patrizio Vaiano, Luigi Zeni, Antonello Cutolo

**Affiliations:** 1Dipartimento di Ingegneria, Università della Campania Luigi Vanvitelli, Via Roma 29, 81031 Aversa, Italy; 2Istituto per il Rilevamento Elettromagnetico dell’Ambiente, Consiglio Nazionale delle Ricerche, Via Diocleziano 328, 81024 Napoli, Italy; 3Dipartimento di Ingegneria, Università degli Studi del Sannio, Corso Garibaldi 107, Palazzo Bosco Lucarelli, 82100 Benevento, Italy; 4Dipartimento di Ingegneria Elettrica e delle Tecnologie dell’Informazione, Università degli Studi di Napoli Federico II, Via Claudio 21, 80125 Napoli, Italy; 5National Institute for Nuclear Physics (INFN), 80125 Napoli, Italy; 6European Organization for Nuclear Research (CERN), CH-1211 Geneva, Switzerland; 7Dipartimento di Ingegneria, Università degli Studi di Napoli Parthenope, Centro Direzionale Isola C4, 80143 Napoli, Italy; 8Optosensing Ltd., Via Carlo de Marco 69, 80137 Napoli, Italy; 9Istituto per i Polimeri, Compositi e Biomateriali Consiglio Nazionale delle Ricerche via Enrico Fermi 1, 80055 Portici, Italy; 10CERICT SCARL, CNOS Center, Viale Traiano, Palazzo ex Poste, 82100 Benevento, Italy; 11Optosmart Ltd., Via Pontano 61, 80122 Napoli, Italy

**Keywords:** optical fiber sensors, fiber Bragg gratings, distributed sensing, infrastructural monitoring, railways safety and security

## Abstract

Our group, involving researchers from different universities in Campania, Italy, has been working for the last twenty years in the field of photonic sensors for safety and security in healthcare, industrial and environment applications. This is the first in a series of three companion papers. In this paper, we introduce the main concepts of the technologies employed for the realization of our photonic sensors. Then, we review our main results concerning the innovative applications for infrastructural and transportation monitoring.

## 1. Introduction

In the last three decades, optical fiber sensors (FOS) have shown their increasing importance in a large variety of applications in the fields of both applied engineering and life science, with particular reference to both safety and security [1,2,3,4,5,6,7,8,9,10,11,12]. This is one of few market sectors with a total world income constantly increasing at a rate higher than 10% per year. As is well known, this is mainly due to the particular properties of the optical fibers, among which are their full electromagnetic compatibility, their small dimensions, their possibility of integrating many sensors on a single optical fiber and their possibility of performing distributed measurements [13,14,15,16,17,18], allowing for the measurement of different parameters.

Our multidisciplinary group has been active in this field for the last twenty years, concentrating most of our attention on the usage of photonics and nanophotonics [19,20,21,22,23,24,25,26,27,28] for both medical and industrial applications [29,30,31,32,33,34,35,36,37,38], by which we have realized a large variety of in-field applications [39,40,41,42,43,44,45,46,47,48,49,50], some of which have reached the market or are operative in some industrial or research plants (e.g., CERN, Geneve, railways) [51,52,53,54,55,56,57,58,59,60,61,62,63].

We have decided to collect, in this paper and in its two companion papers [64,65], our main results concerning both the safety and security in environment, transportation and industrial applications.

Accordingly, this paper is organized as follows. First, we briefly review the basic properties of the optical fiber sensors based on either fiber Bragg gratings (FBGs) or distributed optical effects. Then, we describe a strain sensor based on the use of either bar or QR codes, which is very cheap and easy to use in connection with the Internet of Things (IoT) technologies, although, of course, its characteristics are worse than those of optical fiber sensors. Subsequently, we describe some applications regarding infrastructural monitoring and railways safety. Specifically, we present new systems for measuring the weight of a train based on either point or distributed optical fiber sensors, and verifying if the wheels are perfectly circular or if they have any defects while the train is running. Then, we describe a distributed optical fiber sensor for road traffic monitoring. In the last section, we discuss some innovative systems for anti-intrusion detection in large areas and dangerous gases detection inside tunnels.

## 2. Photonic and Optical Fiber Technology

This section briefly presents the theoretical aspects and mechanisms used in different applications. Here, we discuss the photonic and optical fiber technologies adopted in the proposed applications, including infrastructural and ground transportations monitoring, aerospace and submarine applications [59], agriculture and soil monitoring and sensors for seismic monitoring and for HEP environments [60].

### 2.1. Optical Fiber Grating

Fiber Bragg grating (FBG) is one of the most popular fiber optic devices employed for sensing. Since their first introduction in the late 1970s, FBGs gave rise to several scientific works, commercial products, academic spin-off and private companies [66]. The FBG is developed through a periodic perturbation of the refractive index of the core region of an optical fiber. The perturbation is typically produced by means of UV light passing through an amplitude mask. The FBG periodicity ranges between 500–2000 nm, whereas the typical overall length of the device ranges 1–10 mm. As far as the working principle is concerned, the FBG produces a light coupling between the fundamental core mode and the backward propagating mode. As schematically illustrated in Figure 1a, if a broadband light is injected into the fiber with an FBG, a narrow range of wavelengths (i.e., the Bragg wavelength) will be reflected and the other wavelengths will be transmitted through the fiber. The exact value of the Bragg wavelength is defined by the following phase-matching condition: $\lambda_{B}=2n_{eff}\Lambda$, where $n_{eff}$ is the effective refractive index of the core mode and Λ is the grating period [67]. There different types of FBG, such as chirped or tilted, where the perturbation is, respectively, not uniform or angled with respect to the fiber axis [68]. The effective refractive index and the spatial periodicity of the grating are both affected by changes in strain and temperature. As a result, such devices find significant application for the measurement of several mechanical parameters and more [69]. As the popularity of the FBG sensor grows, so do the various types of sensors that employ the FBG as the sensing element [70]. Nowadays, FBG-based sensors permit a high-sensitive detection of strain, temperature, tilt, acceleration, displacement and so on. They are either currently available or in development for commercial sale [71].

The key feature of the great success of the FBG technology is the high-multiplexing capability enabling quasi-distributed and multi-parameter sensing [72]. The usual multiplexing approach for FBG sensors consists of the wavelength division multiplexing (WDM) technique. By simply writing each FBG sensor at a different central wavelength, each sensor and its operational range can be assigned to a finite spectrum region, preventing the overlapping between adjacent sensors, schematically plotted in Figure 1b. Moreover, alternative approaches have been proposed to enable multiple FBG devices to work in the same spectral range. A first alternative approach is based on time domain multiplexing (TDM), which acquires a signal spread in time from different gratings. Superimposed FBGs have been demonstrated in silica fiber, mainly for code division multiple access (CDMA) applications and dense WDM multiplexing [73].

Long-period gratings (LPGs) couple the power between the guided core mode and the cladding modes and were first proposed in [74]. They were realized by exposing the core of a Ge-doped fiber to a periodic UV pattern. The periodicity of the modulation ranges in hundreds of micrometers (100–1000 μm) and causes the light to couple from the fundamental guided mode to discrete forward-propagating cladding modes [75]. The principle of operation of an LPG is illustrated in Figure 2: if a broadband light is injected at the input of the grating, the resulting output spectrum of the fiber will present discrete resonance bands. Each one is the result of the coupling between the core mode and a different cladding mode and it is located at a wavelength *λ_i_*, satisfying the phase-matching condition $\lambda_{i}=\left(n_{co}-n_{cl,i}\right)\cdot \Lambda$, where $n_{co}$ and $n_{cl,i}$ are, respectively, the effective refractive indices of the core and ith order cladding mode, while Λ is the grating period [76]. For standard optical fibers, the difference between the core and cladding effective refractive indices is typically between 10^−3^ and 10^−2^, thus, for coupling to occur at wavelengths in the communication window (1200–1600 nm), the periods are in hundreds of micrometers. Since cladding modes attenuate rapidly while propagating due to fiber bending and absorption due to the coating, the light at the phase-matched wavelengths is lost from the fiber, while nothing is reflected backwards. It should be also noted that the bands have different attenuation values and bandwidth due to distinct coupling coefficients dependent on modal overlap [77]. LPGs can be fabricated through a period modification of the physical and geometrical properties of an optical fiber. There are several techniques that have been applied over the years for this purpose, for example: irradiation with UV lasers [76], CO_2_ lasers [77], IR femtosecond lasers [78] and application of electric arc discharge [79,80,81]. Depending on the technique, the perturbation can affect the refractive index and/or the geometry of both the core and cladding regions. The sensing mechanism of LPGs can be explained on the basis of their phase-matching condition, revealing that the resonance wavelength is a function of the effective refractive indices of the guided and cladding modes and the grating period [82]. The effective indices are indeed dependent on the fiber’s physical and geometrical parameters, such as the core and cladding refractive indices and radii. External perturbations, like strain and temperature, modulate various optical fiber parameters and the grating period, resulting in changes in the resonance wavelengths. As the spectral variation of the effective indices of the cladding modes are different, the corresponding attenuation bands are also expected to exhibit distinct wavelength shifts. In addition, since the resonance wavelength values are dependent on the grating period and writing conditions, these parameters also contribute. Finally, it is important to highlight that the cladding mode effective refractive index is also dependent upon the refractive index of the medium surrounding the cladding (surrounding refractive index—SRI) [83] and such characteristics are well exploited for the development of biochemical sensors [84,85].

### 2.2. Distributed Optical Fiber Sensors

In the last few decades, distributed optical fiber sensors have emerged as an important tool for the structural health monitoring of bridges, tunnels, aircrafts, etc., as they permit following the variations of temperature and strain with a single conventional optical fiber, with good accuracy and spatial resolution (see Figure 3) [86]. The main parameters qualifying distributed optical fiber sensors are the measurement range, the spatial resolution (i.e., the capability of resolving two close perturbations along the same fiber), the measurand accuracy and the acquisition rate. Typically, for any given technology, these parameters can be traded off through a proper system configuration [87].

Different scattering phenomena provide a mechanism to turn a conventional optical fiber in a distributed sensor. The common mechanisms employed for temperature and strain sensing are the backward Rayleigh and Brillouin scattering and the forward Brillouin scattering. The latter has recently gained interest due to its capability of recovering the acoustic properties of liquids [88]; however, the structural health monitoring field is still dominated by backward Rayleigh and Brillouin scattering-based sensors.

Rayleigh scattering arises from the intrinsic inhomogeneities of the fiber material. Since the early days of optical fibers, the Rayleigh backscattering has been exploited in optical time-domain reflectometry (OTDR) for the remote measurement of fiber loss and fault location [89]. Different from conventional OTDRs, the distributed sensors rely on the Rayleigh scattering make use of highly coherent laser sources, which result in a jagged photo-detected signal strongly depending on the amplitude and phase characteristics of the single scatterers [86]. Any subtle variation of the scatterer’s distribution, induced by a temperature or a strain variation, will vary the amplitude, phase and state of polarization of the detected signal. For quantitative measurements, the acquisition of either the amplitude of the backscatter signal at several laser wavelengths, or the phase of the backscatter signal at a fixed wavelength is usually required. For static measurements, the former approach is usually adopted, often using a wavelength-swept laser [90]. Instead, the latter approach is more suitable for dynamic measurements [91].

Brillouin scattering results from a Bragg-type reflection of the incident light from a moving diffraction grating, created by a propagating acoustic wave through the elasto-optic effect. The acoustic wave can be generated either spontaneously by thermal excitation (spontaneous Brillouin scattering), or optically via electrostriction (stimulated Brillouin scattering). Stimulated Brillouin scattering attains a higher efficiency than spontaneous Brillouin scattering, therefore it is preferred when a high spatial resolution and/or high accuracy are required. In a single-mode optical fiber, the frequency shift between the incident light and the backward scattered light, known as the Brillouin frequency shift (BFS), is given by $BFS=2n_{eff}V_{a}/\lambda$, where $n_{eff}$ is the effective refractive index of the optical mode, $V_{a}$ is the acoustic velocity of the medium and λ is the free-space pump wavelength. As the BFS depends on the effective refractive index and acoustic velocity, it changes whenever these quantities change in response to local environmental variations and therefore can be used to deduce the temperature and strain along the fiber.

When it comes to choosing the most suitable distributed sensing technology for a given application, several factors must be taken into account. A first distinction can be made based on how fast the phenomenon to be monitored evolves: for static or slowly varying phenomena, Brillouin scattering sensors are often preferable, as they permit self-referenced, fading-free measurements (fading is a phenomenon typical of interference-based sensors, such as those based on coherent Rayleigh scattering). Furthermore, they are capable of extremely long measurement ranges (up to one hundred km and more), making them ideally suited, e.g., in pipeline monitoring or in the early warning of ground movements over large areas. The spatial resolution is typically 1 m, even though some special configurations push the resolution down to the cm or even mm range, at the expense of a longer acquisition time [92,93]. Brillouin scattering-based sensors may even obtain high acquisition rates through special configurations [94,95,96], however, they cannot capture weak vibrations at nε-level strain, mainly because of the poor sensitivity of the BFS to strain (∼50 kHz/µε). Therefore, the use of Rayleigh scattering-based sensors is often the preferred approach for dynamic measurements. Numerous dynamic configurations, generally referred to as distributed acoustic sensors (DAS), have been demonstrated. The DAS technology, originally developed for the perimetral control of large areas through optical fiber cables buried underground, now attracts great interest in the field of seismology [97,98], owing to its high sensitivity and the availability of a capillary network of deployed optical fiber cables borrowed by the telecom industry.

Another distinction can be made in terms of fiber access: Rayleigh-based sensors require access to only one end of the optical fiber. Instead, Brillouin-based sensors come in two flavors: those based on stimulated Brillouin scattering, requiring access to both ends of the fiber, and those based on spontaneous Brillouin scattering, requiring access to only one end. Single-ended configurations have the advantage of being fault-tolerant, i.e., they continue to operate even if the fiber breaks at some point. Instead, double-ended configurations fail completely when losing the optical continuity, making their use more critical in harsh environments.

### 2.3. Bar and QR Code-Based Strain Sensors

The increasing interest in the Internet of Things (IOT), the increase of the automation processes and the more pressing requirements for both safety and security continuously enhance the demand for new sensing systems that are cheaper and simpler [99,100,101,102,103,104,105,106]. The possibility of integration with mobile phone and wireless interrogations is becoming more instrumental [107,108,109,110,111,112,113,114,115]. One of the simplest solutions to store some data together with the possibility to retrieve them by simply using a camera of a mobile phone is offered by the well-known bar- and QR codes. In this line of argument, we developed the idea of using either one of them as a simple strain sensor. Two different installing configurations were considered. As it will be shown later, we have foreseen a very simple configuration where the sensor can be simply interrogated by a mobile phone in order to know the strain and, hence, the stress status of the structure where the code has been applied. Then, a slightly more sophisticated (but still very simple and very cheap) configuration is described. In this last case, the sensor is integrated inside a small structure containing a camera, a small antenna, a battery and a microprocessor. This device, which is very cheap and smaller than a cubic centimeter, can be applied everywhere (either in view or embedded) and can automatically transmit the strain information. The first one is indicated more for periodical control of the structure, while the second configuration is indicated when continuous monitoring is required. A rough money estimate reveals that, by this approach, it becomes possible to monitor the health state of either a building or a bridge with more than one hundred sensors and a small Internet-connected PC with a total price less than a few thousand euros. In the last section, some preliminary results are discussed as well. We explicitly underline that both the precision and the sensitivity available with this kind of sensor are worse than those available with other kinds of sensors (e.g., optical fiber sensors). In fact, the best sensitivity of our sensors is never better than a few micro strains, where 0.1 micro strain can be easily attained by an optical fiber sensor. We explicitly remind readers that, in many structural and mechanical applications, a strain sensitivity of the order of 100 microstrains (equivalent to one millimeter over 10 m) is already good for a satisfactory monitoring process. The best advantages of the proposed sensors are the low price, the easy handling and the wireless use, thus making them the best candidate for structural health monitoring in the IoT philosophy. Our idea is based on a new strain sensor based on the use of either barcodes or QR codes. The sensor is wireless and temperature-compensated. It can be mounted either embedded (also in buildings of historical and artistic importance) or on sight. In addition, depending on the particular configuration, it can be interfaced with either a PC or a mobile phone [113].

The operating principle is quite simple, and it involves the application of a barcode or QR codes on the surface to be monitored (STBM). Indeed, even if the analysis will be performed only by scanning a barcode, it will be clear that our sensor can work with any kind of image. Hereafter, by barcode, we include either the QR code or the barcode itself. After the barcode has been applied on the STBM, its image is stored by using a mobile phone or a camera. By observer, we shall mean either one of these two devices. After the barcode has been applied on the STBM, its image is stored by the observer, and it is considered the reference of the initial conditions of the STBM. By SI, we mean this initial stored image. At a different time, we again take a picture of the applied barcode, and we call this image the actual image (AI). Of course, if the STBM will experience some deformations, even the barcode will be deformed, thus resulting in a difference between the AI and SI, from which a measurement of the STBM deformation can be derived according to the analytical model reported in the next section. The comparison between the two images (stored and actual) is made with an image processing program that transforms the analog image (acquired) into a digital image. This digital image is elaborated and transformed in a signal through which the information of deformation and other information that needs to be measured is extracted. The simplest operational configuration involves the applied image mounted on the surface to be examined and its inspection is performed with a simple cell phone, inside which the stored image has been loaded together with an application capable of comparing the two images and providing a quantitative measure of any deformation observed (Figure 4).

The use of a mobile phone is the cheapest method, but it demands an operator that occasionally interrogates the barcode. In order to automate the monitoring process of a structure, a camera is required. In this case, the camera must be equipped with a mini circuit with a memory accompanied by a suitable acquisition and processing system. To this aim, we have designed an active configuration in which the device consists of a parallelepiped of transparent material (Figure 5). On one face, we put the applied image (Figure 5 point 4), while on the opposite face, the camera (Figure 5 point 1) is mounted with the acquisition and processing circuit (Figure 5 point 2). The system can be completed with a light source (e.g., an LED) to illuminate the space between the applied image and the camera. The system can be equipped with a mini battery and a mini antenna to transmit the information to a control unit (Figure 5 point 5).

We discuss the theoretical model of our device for a better understanding of its basic principle. Even if our device can be used with any kind of figure, QR Code or barcode, here we refer to a very simple configuration that can be analyzed with a very simple analytical model by which we can easily estimate the main performance.

With reference to Figure 6, we consider a barcode where the width of the white lines (W) and black lines (B) are exact equal to d/2. The barcode is a periodic figure with its period equal to d and its length is equal to L = Nd, where N is the number of periods.

Now we discuss the theoretical model of our sensor, referring to a very simplified model that allows us to use a very simplified analytic model that is very useful for understanding the basic principle of our device. We explicitly underline that our device can work with any kind of figure different from that which we are analyzing here. In the barcode above, we have chosen, in the starting configuration, the thicknesses of the black and white lines perfectly equal to each other. More specifically, just after the device has been installed on the structure to be controlled, we have
(1)$${\left.\frac{B}{B+W}\right|}_{t=0}=0.5$$
where *B* and *W* are the width of the black and white lines, respectively. This amounts to saying that AI = SI in the initial instant of time. As AI is equal to SI, when we superimpose the two figures, we can see that the total white area is equal to the total black area. When, after a given period of time, the STBM experiences a deformation (e.g., it becomes longer), the AI becomes different from SI. If we obtain an image that shows the deformation of the AI is normal to the lines of the barcode, then we get the situation shown in Figure 7, where both the initial and deformed barcode lines are shown.

If we superimpose the two images, the resulting width of both the white and black lines will change. In the barcode with no deformation (SI), we have = *d*/2, … = *nd*/2 and the total length is equal to *Md*, where *M* is the number of periods of the barcode. If the AI has no changes, we have
(2)$$B_{T}=W_{T}=Md/2$$
(3)$$\frac{B_{T}}{W_{T}+B_{T}}=1/2$$
where *B_T_* and *W_T_* are the dimensions of the total black and white lines, respectively.

After the barcode has been deformed to an amount equal to ε, the new abscise $x_{1}{}^{\prime }$ will be equal to
(4)$$x_{n}{}^{\prime }=\frac{d}{2}\cdot n\left(1+\varepsilon \right)$$

On the other hand, when the IA changes, as in Figure 7, after superposing IA with IM, we get:(5)$$B_{T}=\frac{M\cdot d}{2}+\overset{M}{\underset{n=1}{\sum }}n\cdot \varepsilon \cdot \frac{d}{2}=\frac{M\cdot t}{2}+\frac{M\cdot \left(M+1\right)}{2}\cdot \varepsilon \cdot \frac{d}{2}$$
(6)$$W_{T}=\frac{M\cdot d}{2}-\frac{M\cdot \left(M+1\right)}{2}\cdot \varepsilon \cdot \frac{d}{2}$$
(7)$$\frac{W_{T}}{W_{T}+B_{T}}=\frac{\frac{M\cdot t}{2}-\frac{M\cdot \left(M+1\right)}{2}\cdot \varepsilon \cdot \frac{d}{2}}{M_{t}}$$
(8)$$\frac{W_{T}}{W_{T}+B_{T}}=\frac{1}{2}-\frac{M+1}{2}\cdot \varepsilon$$
(9)$$\frac{B_{T}}{W_{T}+B_{T}}=\frac{1}{2}+\frac{M+1}{2}\cdot \varepsilon$$

As matter of fact, Equations (8) and (9) are the characteristic equations of our sensor.

Some preliminary measurements have demonstrated that our model is correct and that, with no special precautions, the sensitivity is better that 1 mm over 10 m, which is sufficient for civil engineering monitoring. Our model also provides the benefit of a very low price of the proposed sensors. We think that this device is extremely useful for easy monitoring of buildings, bridges and industrial plants, including the fatigue control of mechanical pieces.

## 3. Railway and Road Transportation

### 3.1. WIM and WILD System Based on FBG Sensor for Railway Safety

Rail transport has grown significantly in the last several years for both passengers and goods: for this reason, the need to increase the levels of traffic safety and security, as well as an efficient and economical maintenance policy, are the most important goals for the railway industry [116]. In this scenario, the smart monitoring systems able to retrieve information on the rails and on the trains during their normal operation play an important role for railway safety, as well as maintenance optimization. Currently, the railway industry uses various detection technologies in order to inspect and monitor rolling stock health status, particularly in terms of wheel- and truck-related irregularities. The possible approaches can be divided in two main groups: vehicle-based or on-board approaches, and wayside or track-based methods [117].

In the first case, the sensor networks must be installed in all the vehicles, and the systems are complex in terms of mounting, implementation and maintenance. In the second case, instead, the sensors are mounted on the rail or along it, and this type of system returns diagnostic information on rolling stocks passing in the instrumented section and on the rail line, avoiding the installation of devices in every vehicle.

By measuring the vertical force generated by the wheel–rail interaction during the normal operation of the trains and the related dynamic track response, information on the quality status of the single wheels, as well as the distribution of weight on the wagons, axles and wheels during normal traffic operation, can be obtained. These kinds of systems are widely used, and they are known as WILD (wheel impact load measurement) and WIM (weight in motion) systems, respectively [118]. The WILD system is used to identify wheels with potential tread defects, such as flat spots, out-of-rounds, built-up treads and shells that result in high impact loads, which cause damage to the vehicle and truck components, as well as to the track structure. Instead, the WIM system is of great interest in controlling axles and vehicle loads, detecting overloads and unbalances and predicting train derailment in order to guarantee acceptable train operations and safety.

Usually, WIM systems are based on rail strain measurements that use strain gauges installed on the rails to obtain signals that are proportional to the applied loads [119], while the WILD systems use both acceleration and strain gauge sensors to measure the contact force and to detect defected wheels. Although most applications report the use of conventional technologies as electrical strain gauges [120] or piezoelectric sensors [121], many works using fiber optic sensors for railway applications have been reported thus far [122]. The conventional technologies are well-known and consolidated, but are also inefficient for railway systems since they can be adversely affected by electromagnetic interferences; this problem is completely eliminated by using optical fiber sensors, which provide intrinsic immunity to electromagnetic noise [1,123,124].

For this reason, and due to their inherent distinctive advantages, in the last two decades, a significant number of innovative sensing technologies based on fiber optic sensors have proved to be a powerful tool for meticulous assessment of railway systems, including train and track behaviour, by enabling real-time data collection and inspection and detection of structural degradation [125]. Wei et al. [126] showed that a single FBG glued to the rail can provide useful information about the occupation state, the train composition (through axle counting and weighing in motion), its velocity and acceleration. Filograno et al. [127] installed different FBGs for monitoring railway traffic in Madrid, achieving the detection of flat wheels. Mi et al. [128] utilized an FBG sensor array to monitor the brakes, axle speed and loading conditions of trains. Allotta et al. proposed an algorithm for FBG sensors to estimate train speed [129], and Taheri S. et al. [130] showed the results of a feasibility study conducted to develop new and more advanced sensory systems based on fiber optic sensors, as well as signal processing algorithms capable of detecting various rail surface irregularities. Also, Xiao-Zhou Liu et al. [131] developed a wayside FBG-based wheel condition monitoring system that can identify wheel tread defects online during train passage.

In this framework, we first proposed an optimized package allowing the fastening of the sensor to the foot of the rail, which offers rapid and non-invasive installation, preventing the need to drill the rail [132]. Afterwards, we designed, developed, in-field validated and overall engineered an optical fiber-based sensing system that relies on the use of FBGs’ strain sensors suitably clamped to the railway tracks through suitable and certified metallic packages. The proposed solution combines both the WIM and WILD functionalities, thanks to a judicious signal processing able to extract from the Bragg wavelength shift (induced by travelling wheels) of the single sensors the static contribution related to the wheel weight, and the dynamic sensor that carries information of the healthy status of the wheel itself [133].

The developed system is the result of an efficient synergy work, which has involved a multidisciplinary team composed by the Electronic and Photonic research group of the University of Sannio, the Optosmart Srl and the Hitachi Rail STS company, which currently has included it in its product portfolio. Here, we report on the main characteristics of the system by focusing the attention on the sensor packaging and the installation kit, the operational principle of the single optical strain sensor during the transit of trains and the data processing developed for WIM and WILD outputs. In addition, we show in-field results returned by one of many systems installed in real scenarios, demonstrating its high performance and its potentiality to increase the safety and to support new methodologies for smart maintenance in the railway industry.

#### 3.1.1. Main Characteristics of the WIM and WILD Systems

The WIM–WILD system is a smart solution entirely based on optical fiber technology and fiber optic sensors and combines the two functionalities by using optical fiber strain gauges based on FBG technology. The sensors are integrated in metallic packages and are clamped to the rail foot between two sleepers by a customized stainless-steel package, designed for detecting the vertical forces generated by the single wheel/rail contact in order to measure both quasi-static and dynamic solicitations imposed on the rail during normal transit.

Figure 8 provides the rendering of the sensor and the sensor as it appears after installation under the rail foot (between two adjacent sleepers). On the left, the sensor that is substantially composed by the fiber optic-based sensitive element, visible in the central area, is tightly screwed on a structure designed to install the sensor under the rail base and protect the optical transducer inside, as well. On the right, the sensor is installed under the rail: most of the device body disappears under the rail, in the section between two sleepers.

Thanks to the proposed design, the sensor installation is easy and fast: they can be easily mounted on the rail in a few minutes, during normal train service and without any interruption of the traffic. Furthermore, the maintenance and replacement of the sensors can be carried out without interfering with normal train service, and a single sensor can be removed from the rail and put alongside the track for maintenance/replacement without the removal the other sensors. It is designed for the UIC60 and UNI50 rails, but can be customized upon request for any kind of rail.

Over the years, different installations in real scenarios have been carried out, and an example of an “in-field” system installation carried out in a Milano subway—San Siro station (UNI50 rail type) is shown in Figure 9a: the sensors’ network is a symmetrical distribution of sensing devices and, in this case, there are seven sensors on each side, spaced about 0.6 m apart (the single one installed in the central point of the rail, between two adjacent sleepers), cooperating to measure each passing wheel with the better possible accuracy. The installation layout can be easily customized on the basis of specific site requirements or vehicle characteristics. In Figure 9b, we report a typical raw signal returned by one FBG-packaged sensor installed in Milano, without any kind of post-processing operations (acquired with an interrogation module with 2.5 kHz of frequency sample and repeatability less than 1 pm): at this site, all the (passenger) trains are composed by ten total axles (five carriages). As shown, during the train transit, the single FBC returns a sensorgram characterized by different peaks, each one related to a single wheel.

The total number of peaks is the same of the total number of axles: for this reason, the system also performs as the axles counting measurement. Moreover, by measuring the time delay between two pulses, and knowing the wheelbase distance, the speed is calculated. The amplitude of the single peak retrieves information about the weight of the associated train wheel and, therefore, the distribution weight per wheel, axle and wagon. The unbalance or the overload conditions are also carried out by the system.

From a mathematical point of view, the single Bragg wavelength shift ∆*λ* from a passing train can be defined as follow [134]:(10)Δλ=Δλε+ΔλT=Δλε≈0.78⋅ε

As is well known [134,135,136], ∆*λ* depends on temperature and strain variations (∆*λT* and ∆*λε*, respectively): during the passing of a train, the temperature variation is close to zero, as demonstrated by the baseline signals before and after the convoy’s transit, which are on the same level. For this reason, the Bragg wavelength shift is approximately dependent only on the strains measured under the rail foot and returned by the different wheels in movement on the instrumented rails. Thanks to a judicious tool, at the end of installation and on demand, the single FBGs are calibrated using vehicles of known weight (e.g., engines or test trains), obtaining the sensitivity coefficients α [kg/pm] used to convert the strain into impact force and/or weight measurements [133]:(11)P[kg]=α⋅Δλε⇒F[kN]=g⋅P
where *P* is the load applied on the wheel and g is the gravity acceleration (9.81 m/s^2^). The sensitivity coefficients of the FBGs depend on the tracks (UNI50, UIC60, etc.) and the status of the rail in the installation points: for example, in a Milano subway (rail type: UNI50), the coefficients were, on average, equal to 7 kg/pm, which indicates 70 N/pm as the sensitivity in force.

The raw signal returned by the single FBG contains both quasi-static and dynamic information: the static Bragg wavelength shift (correlated to quasi-static strain induced by a travelling wheel on the track) is strictly related to the wheel weight, while the dynamic Bragg wavelength shift (correlated to the dynamic strain/vibration induced by the traveling wheel) carries important information on the healthy status of the wheel itself. For this reason, the data processing developed for WIM and WILD measurements divides the ∆*λ_ε_* in two parts, as follows [133]:(12)$$\Delta \lambda_{\varepsilon }=\Delta \lambda_{P}+\Delta \lambda_{D}=\Delta \lambda_{LPF}+(\Delta \lambda_{\varepsilon }-\Delta \lambda_{LPF})$$
where ∆*λ_P_* and ∆*λ_D_* are, respectively, the quasi-static and dynamic component on the raw Bragg wavelength shift, the first obtained with a low pass filter. By converting the wavelength shift in loads, the quasi-static and dynamic forces, respectively related to the weights and the quality status of the rolling surface of the single wheels, are obtained as follows [122]:(13)$$\begin{array}{l} F_{P}[kN]=g\cdot \alpha \cdot \Delta \lambda_{P},\\ F_{D}[kN]=g\cdot \alpha \cdot \Delta \lambda_{D}. \end{array}$$

Figure 10a,b reports the quasi-static and dynamic components returned by the single WIM–WILD sensor clamped to the rail during the transit of a two-axle wagon, respectively.

In the case of a defected wheel, a spike appears on the dynamic signal as a result of the impact between the defect and the rail during the normal passing. Therefore, it is clear that the developed system performs all the measurements by using the same sensors and the same hardware installed along the rails: this is a strong advantage with respect to the other competitor systems [122,137], which usually use strain sensors for load outputs and accelerometers for dynamic information. In addition to the WIM outputs, the system provides information in real-time and for the single wheels different in force measures, and it is able to detect defected wheels with high reliability thanks to a dimensionless indicator of the vibrations produced by the single wheel (the so-called defect indicator), obtained by normalizing the dynamic component to the weight and transit speed.

The system is equipped with auto-calibration features combined with a full auto-diagnostic architecture able to continuously check the correctness of the measurements and the integrity of the different components of the system. In addition, the system redundancy is implemented to allow correct operation even in the case of 50% sensor failures for both WIM and WILD outputs.

#### 3.1.2. In-field Results

In 2007, the Electronic and Photonic research group of the University del Sannio and the Optosmart Srl, have signed an agreement with Hitachi Rail STS (originally Ansaldo STS, a company of the Finmeccanica group) for the development of trackside monitoring systems based on the use of fiber optic sensors to improve safety and security levels of the national and international transportation system.

Over the years, thanks to this tight collaboration, different installations and experimental in-field trials have been carried out in various railway scenarios (in both UNI50 and UIC60 rail types), where the trains pass in normal operation (just to name few: Bolzano, Genoa, Marcianise, Naples and Milan), including the one realized in 2014 in Napoli central station, thanks to activities included in a research project PON 01_00142 “Sicurfer” financed by the Italian Ministry of Research (MIUR) [133], which allowed us to validate the WIM–WILD system both from a functional and performance point of view, and to bring it to fruition. Since February 2015, several systems have been sold by Hitachi Rail STS and are currently running in different international sites, such as, for instance, the case of the new railway line in Abu Dhabi (United Arab Emirates, Ruwais and Shah sites), or the system installed in a Milan subway—San Siro station.

In the following subsection, we report, as examples, some WIM and WILD results in order to demonstrate the capability of the developed system to perform with high accuracy (typically less than 2%) the load measurements and the wheel defect identification.

#### 3.1.3. WIM Results

Here we show the results returned by different in-field systems installed and actively running. In Figure 11a, we report the typical waveform returned by one WIM–WILD sensor installed in the United Arab Emirates (Abu Dhabi site) and filtered with an LPF filter in order to measure the load applied on each wheel. The passed train is composed of two engines, one at the head and one at the tail, and different undercarriage wagons composed of two carriages and four axles. The peaks returned by the engines are greater than those related to the wagon wheels, and this agrees with the real loads: the engines are the heaviest wagons because they contain the motor and the fuel loads.

Figure 11b shows the WIM outputs carried out by the same system for sixty undercarriage wagons, all together freight wagons, each one with the same total nominal weight (28,600 kg), passed in empty condition on the instrumented rail. By using these results, we calculated the medium estimated load and the WIM accuracy (as the ratio between the medium error and the nominal weight, in percent), obtaining a value very close to zero (0.4%).

Lastly, in Figure 11c, we show the WIM results returned by the WIM–WILD system installed in a Milan subway (San Siro Station) for 290 passenger trains, all together with a nominal empty total load of approx. 78 tons. In this case, by subtracting the nominal empty weight from the estimated total weight, and assigning a load of 70 kg per person, the total number of passengers is calculated.

#### 3.1.4. WILD Results

For this kind of measurement, in order to demonstrate the high potentiality of the developed system for defected wheels detection, we report the results obtained during experimental trials carried out in a Marcianise goods station (UIC60 rail type). Here, a test train with a healthy wheel and a flat wheel (with a length of 39 mm and 76 mm) passed over the instrumented rail, and the system, for the single under test wheel, returned a dynamic impact force, as shown in Figure 12 (with a speed of 30 km/h). For WILD measurements, the single strain sensor is used to control a small portion of the rolling surface of each wheel (about 0.6 m) and the reconstruction of the entire circumference in terms of the dynamic impact force is given by a judicious data fusion tool which, merges the single responses.

In the Marcianise installation, we used five sensors per side to cover three meters of the rolling surface, but this number can be extended on demand to cover different wheel envelopes (for example, in Milan, the subway is equal to 7). By returning the dynamic signals as a function of the flat length, the capability of the system to detect a defect and to estimate its size is clear: for new wheels, in fact, there are no peaks on the dynamic signal; instead, for flat wheels, a pulse with an amplitude proportional to the flat length is evident. Other reasonable results are returned by the system installed in Napoli central station [133], where we have obtained a full agreement between the system outputs and the measurements returned by a scanner laser probe for the single wheels of a test train.

In the Milan subway, we acquired the impact forces and the different dynamic observables produced by almost six thousand wheels. For all the selected wheels, we carried out an analysis on the vibration patterns (defect indicator) in order to identify healthy (Figure 13b) and outlier wheels with high vibration levels (Figure 13c). Only 54 of 5800 measured wheels returned an out-of-statistical defect indicator, and none of them seemed to have a flat-type defect, as the vibrations were distributed over the entire circumference of the transited wheel. In Figure 13a, for example, we report the maximum defect indicator values recorded for two wheels of 290 transited trains. For all the outlier wheels (with a maximum defect indicator value greater than 60), the system launched an alert in real time, identifying the anomalous condition.

Here we report a brief description of trackside monitoring systems based on the use of fiber optic sensors to improve safety and security levels of railway traffic, developed by the Electronic and Photonic research group of the University del Sannio in close collaboration with Hitachi Rail STS and Optosmart Srl companies. The diagnostic system performs the WIM and WILD functionalities in order to measure the loads applied on the single wheels and to identify defected wheels in real-time. During the description, we showed different in-field results carried out in different installations (for both UNI50 and UIC 60 rail types), demonstrating the correct operation, the high WIM accuracy (less than 2%) and the high reliability in the detection of defected wheels. In other words, we demonstrated the capability of the proposed tool to support new methodologies for enhancing the safety levels in railway traffic.

### 3.2. Railway Traffic Monitoring and WIM Based on Distributed Optical Fiber Sensors

In railway infrastructure monitoring, FBGs have been increasingly used thanks to their numerous advantages over conventional sensors [123,138]. On the other hand, distributed optical sensors may further increase the level of safety and add more complete insight into the rail behavior under the train passage, thanks to the wealth of information that they offer. In the following, we report the application of a distributed optical fiber sensor based on Brillouin scattering for train identification and dynamic load measurement [139]. The rail sector chosen for our monitoring campaign is a continuous welded rail situated near the San Menaio station, around kilometer post (KP) 6 of the San Severo–Peschici regional line (Italy). A length of 60 m of single-mode optical fiber was glued along the rail at the position indicated in Figure 14a. Field tests were carried out by a portable prototype based on the Brillouin optical time-domain analysis (BOTDA), operated in a slope-assisted configuration. In such a configuration, the frequency shift between the pump and probe waves is kept fixed at a point lying along the Brillouin gain spectrum (BGS) slope of the fiber. The strain changes are then detected through the induced modulation of the transmitted probe intensity [94].

We report in Figure 14b the raw data acquired by the optical fiber sensor during a temporal interval including a train passage event. The footprint left by train passage is visible as a series of diagonal lines, revealing the passage of eight train axles. The non-uniform image background is due to the static strain-related Brillouin frequency shift spatial variations, resulting in a DC gain varying along the fiber. To better illustrate this point, the Brillouin shift static profile is superimposed on the dynamic measurements in Figure 14b. Despite the limited SNR, the data shown in Figure 14b can be processed in order to extract several features regarding the passing train, such as the number of axles, the axle distance, the train speed and the train weight [139]. We report in Figure 15 the axle distances extracted by the acquired data by identifying, for each position, the instant at which a strain peak is present. In doing so, the diagonal traces associated to each train axle are clearly recognized. The extracted axle distances match well with the nominal values, with a maximum deviation less than 2%.

While tracking the strain peaks permits us to easily recover the axle distance and the train speed, the estimate of weight (“weight in motion”) is more complex. In our tests, we have adopted the Winkler model [139], according to which, the vertical deflection of the rail due to a load *P* imposed on *z* = 0 can be expressed as [140]:(14)$$\eta \left(z\right)=P\frac{\lambda }{2bk_{n}}e^{-\lambda z}\left(sin\lambda z+cos\lambda z\right)$$
where *z* is the coordinate along the rail, *b* is the sleeper width, $k_{n}$ is the rail bed modulus and *λ* is a characteristic length given by $\lambda =\sqrt[{4}]{bk_{n}/4EJ}$, where *E* and *J* are the Young’s modulus and inertial momentum of the rail, respectively. From Equation (14), the corresponding strain profile can be easily derived through differentiation. Comparing the theoretical and experimental strain profile, the weight of each axle can be properly estimated [139]. It is important to observe that applying the model expressed by Equation (14) over the strain distributions provided by the sensor not only provides an estimate of the train weight, but also estimates the bed modulus $k_{n}$, which has a direct relationship with the performance level, rail track safety and amount of needed repair and maintenance [141].

### 3.3. Road Traffic Monitoring Based on Distributed Optical Fiber Sensors

Optical fiber sensors have recently been increasingly applied for the detection of seismic vibrations, using an emerging technology known as distributed acoustic sensing (DAS). Thanks to DAS systems, the optical fiber acts as a spatially continuous array of microphones, where each section of the fiber operates as an independent channel [142]. This capability opens a new range of applications, which would be impossible to even imagine with conventional distributed sensing techniques. One of the main advantages of DAS systems is that the sensing optical fiber is not required to be fixed to the structure to be monitored, as it can even detect acoustic waves excited at some distance. Based on this concept, the DAS technology can be applied for automated traffic monitoring, taking advantage of the vibrations excited by the traveling vehicles.

Road traffic monitoring is usually realized using loop detectors, ultrasonic sensors or cameras. While video cameras are effective in achieving precise vehicle tracking and classification [143], they suffer from high power consumption and can function only with proper illumination. Other technologies include magnetic sensors, radar sensors and microwave detectors, which are limited in terms of the number of sensing positions. Here, we present a method for traffic detection based on the use of a DAS sensor realizing a phase-sensitive, optical time-domain reflectometry (ϕ-OTDR) configuration. The sensor output was processed in real-time using the Hough transform (HT), a feature extraction technique usually employed to identify a certain class of shapes in images [144]. DAS measurements were carried out using a custom prototype, measuring the intensity of the Rayleigh backscattered light excited by an excitation light pulse. The concept is very similar to conventional optical time-domain reflectometry (OTDR), with the fundamental difference that a laser with high coherence is employed as the optical source. The coherence of the laser results in a strong interference between the backscattered lights occurring along the portion of fiber occupied by the pulse. As a result, the backscattered light, measured as a function of time from the instant of launch of the pulse, has a typical speckled appearance. When a vibration, such as the one generated by a vehicle circulating above the buried fiber reaches the fiber itself, the recorded speckle changes its pattern, making it possible to detect and track each vehicle in real-time. Furthermore, by analyzing these vibrations, the vehicle can be roughly characterized, as shown later.

An optical fiber placed under a two-lane two-way state road at a depth of about 40 cm under the road surface was selected for our experiments. The acquired waveforms were grouped in frames, each one corresponding to 30 s of continuous recording. As the vibration produced by each vehicle passage appears as a line segment in the data frame, the Hough transform can be efficiently applied to detect these lines automatically and infer some information about the vehicle, such as its position, speed or class [144].

In Figure 16, we show an example of application of the HT for line detection in an acquired frame. In this case, four vehicle passages were detected by the HT. In order to determine the capabilities of our sensor to monitor the traffic flow for a determined road segment, the ϕ-OTDR sensor was operated continuously for 60 min. During this time, the road length was also monitored through video recording. The recorded video was then analyzed offline in order to determine the number of vehicle passages over the lane situated above the buried duct and compare this data with those extracted from the DAS measurements. Figure 17 compares the actual passages with the HT-determined passages during each 10-min frame. The detected passages are a fraction of the actual ones, with an overall detection success rate of 73% (456 detected passages out of 623). The detection rate in our tests was mostly limited by the signal-to-noise ratio (SNR) of the measurements. In fact, the trace impressed by the passage of the light vehicles was sometimes too weak to be detected by the HT. Using an optimized sensor or a more efficient preprocessing of the acquired data [145] may significantly improve the success rate.

### 3.4. Gas Monitoring in Railway Tunnels Based on LPG Sensors

The ever-increasing demand of hydrocarbons for fuel applications in turn increases their transportation frequency on both rails and roads. As hydrocarbons have highly flammable and volatile behavior even at low concentrations, a real-time monitoring of leaks during their transportation is of vital importance for both environmental and public health. To date, there is a gap in the use of commercial gas sensors, as they require high operating temperatures and are hardly suitable for harsh, hostile and remote environments [146]. An important candidate for this purpose is the fiber optic technology due to its several advantages, such as being intrinsically safe, immune to electromagnetic interferences, easy multiplexing and long-distance monitoring, as well as the possibility to detect a wide plethora of chemical and gaseous species [84].

A monitoring system for the detection of liquefied petroleum gas in railway tunnels based on fiber technology was demonstrated in [147], under the Italian project OPTOFER “Innovative optoelectronic technologies for the monitoring and diagnostics of the railway infrastructure”. The system was deployed into a ~1 km long tunnel connecting two stations in Benevento, along the line Avellino–Benevento run by Italian Railways. The acquisition system was kept inside the station together with the other rail equipment. The sensing technology is based on an LPG coated with a sensitive layer of atactic polystyrene (aPS) for the gas detection, whereas FBG sensors were employed for the measurement of environmental parameters, such as temperature and humidity [148]. The fiber optic sensors were mounted inside a metallic package to enable their installation in tunnel harsh conditions. The monitoring device was finally tested in different conditions, i.e., while the railway line was under normal operation with trains in transit and under controlled gas leakages.

The core of the system is based on an LPG sensor due to its favorable properties for the development of chemical and biological sensors [27,28,85]. The device was fabricated into a standard single-mode fiber and the period Λ was selected equal to 360 μm in order to have the coupling with a high order cladding mode in the near-infrared wavelength range with high sensitivity. In addition, a single-ended (or reflection) probe was developed starting from the LPG to eliminate mechanical cross-sensitivities. With the aim of providing the grating with chemical sensitivity and selectivity, the fiber with the sensor was coated with a nano-scale layer of aPS due to the affinity between its olefin chains and hydrocarbon molecules. Moreover, the thickness of the aPS film was carefully designed through numerical modeling in order to enhance the sensitivity of the LPG by means of the mode transition phenomenon [149,150,151]. The LPG spectra during the different fabrication phases are reported in Figure 18a, highlighting the spectral features with high visibility and low noise. The resulting device presented high sensitivity, selectivity and simple handling. The response of the sensor was first characterized in controlled laboratory conditions by exposing it to butane (which is the main component of liquified petroleum gas) mixtures at very low concentrations in the range 0.1–1.0 vol%, i.e., well below its lower explosive limit (LEL). The results of the characterization, reporting the gas concentration as a function of the LPG resonance wavelength shift, are illustrated in Figure 18b. Fitting of the experimental data was obtained with a 3rd order polynomial function, highlighting that the sensitivity increases with the butane concentration. Fast response time and good reversibility of the sensor were also obtained [152].

The gas sensor was embedded into a shielding case together with an array of FBGs for the compensation of environmental parameters and mounted inside the tunnel. Connecting optical fibers permitted us to measure the sensor responses from a remote location inside a railway station. The data were acquired using a PC with custom software for the elaboration. A schematic and photos of the monitoring system developed are reported in Figure 19.

Figure 20 illustrates an exemplary testing session, where the gas sensor response is reported, as well as the measurement of temperature and humidity inside the tunnel. The test lasted approximately 24 h and included three main moments: two controlled gas leakages (red boxes) and normal operation of the line with train transits (blue box). Events associated with trains and gas expositions are marked in the picture for the sake of clarity. Figure 20a shows the raw wavelength shift acquired for the gas sensor and compensated response, obtained by removing the contributions due to temperature and humidity changes (as measured from Figure 20b). Focusing the attention on controlled gas leakages, reported with red boxes at the beginning of Day 1 and Day 2, it is possible to observe that gas presence produced sudden blue shifts in the resonance wavelength in the raw data and, of course, in the compensated data, as well. A good reversibility can also be observed as the gas was dispersed in the air after injection. On the other side, the long-term monitoring reported in the blue box (without gas leakages but with train transits) illustrates a drift in the raw response of the sensor due to changing of environmental conditions. However, such drift is well eliminated in the compensated response, together with the events associated with train transits.

As an overall assessment, it was always possible to detect gas traces during controlled expositions. Moreover, during normal operation of the railway line, the readings associated with false positives were below 10% of the gas LEL. Finally, it should be remarked that testing was conducted in an unfavorable scenario with diesel trains and errors could probably be reduced in the case of electrical trains.

### 3.5. Intrusion Detection Systems for the Protection of Railway Assets

In this section, we present a brief survey on recent progress in fiber optic sensors for the development of intrusion detection systems. The protection of large areas concerns the detection of accesses or unauthorized activities in several applicative scenarios ranging from airports surveillances, railways assets protection up to restricted-access zones in private properties. Nonetheless, the definition of the potential threats and the specific operative conditions related to the application field can strongly affect the characteristics of the identified technological solution. In this regard, the protection of railways assets represents specific requirements and challenges. Some of the concerns associated with the protection of a railway area is related to the unavoidable presence of the railroad tracks, which cannot be obstructed with physical barriers, and to the electromagnetic interferences associated with the train transit. Then, we specifically focus on research activities carried out at the Optoelectronic Division—Engineering Department of the University of Sannio, in collaboration with Ansaldo-STS (now Hitachi) and Ente Autonomo Volturno (EAV), a local railways operator, devoted to the development of a fiber optic intrusion detection system for the protection of railways assets.

In the few last decades, to meet the increasing demand for security, worldwide research on intrusion detection sensing systems has grown significantly [153]. The recent bombing of the Nord Stream pipeline highlighted how important the protection of strategic infrastructures extending to large areas is. Governments and public and private entities need solutions now more than ever to provide intrusion detection, intelligent surveillance and access to reporting in strategic areas.

Commercially available intrusion detection systems make use of several transduction principles (e.g., vibration sensing, infrared devices, magnetic field, videocameras, etc.), which typically imply different strength and weakness points [153]. The synergy between different and complementary sensing technologies can offer information redundancy and improved reliability.

Despite the necessity of using several solutions based on different technologies, there is still minimal literature available regarding the use of optical fiber sensing technologies for intrusion detection. Nonetheless, compared to conventional technologies, optical fiber sensing technology is potentially well-suited for protecting large areas from unauthorized access with all the additional desirable attributes connected to the fiber optic sensing solutions. In fact, it is well recognized that optical fiber sensors offer specific advantages for advanced security systems because they are small, lightweight, environmentally rugged and having increased sensitivity with respect to traditional sensing techniques [48,154]. Additionally, the optical fiber itself can act also as a transmission medium by strongly simplifying the telemetry system.

Allwood et al. [9] have recently reviewed the fiber optic techniques used in physical intrusion detection systems. They envisaged fundamentally three optical fiber technological approaches: interferometry, scattering and FBGs-based detection. Sensing approaches based on interferometric schemes, such as Sagnac loops, Michelson or dual arms Mach–Zehnder interferometers [9], as well as speckle pattern configurations [155,156], were proposed. They provided high sensitivities, but also a marked sensitivity to weather conditions and to other external disturbance factors. Evidently, the efficacy of an intrusion detection system must be carefully balanced with the warning messages’ reliability as far as ultrasensitive detection systems can incur in false alarms.

Optical fiber scattering techniques have also been proposed for distributed sensing [87,157], demonstrating their ability to be suitable to cover large distances, although requiring quite expensive interrogation equipment and a moderate installation complexity.

FBG-based systems were also successfully used in security applications for monitoring access points, such as windows and doors [158] or in fence perimeter systems [159].

In the following, we report on the development and field test of a fiber optic sensor for an intrusion detection system in a railway asset. Railway infrastructures represent the transportation system able to underpin the growing demand in transferring both cargo and passengers. In spite of their strategic role in the economy and in daily human life, railways are particularly vulnerable to vandalism and intentional damages. Transport operators have growing concerns about the safety and security of railway facilities, workers and passengers because of the intrinsic difficulty in surveilling all railways assets [125,160]. Specific issues concerning the railway transport field relies on vandalism and theft. Railway vandalism can include a wide range of criminal activities, including drawing graffiti and damaging trains and other company properties. It may be surprising, but transport infrastructures become frequently severely disrupted because of theft of copper cabling along the line.

In order to avoid service interruptions and economical losses, efficient security systems in railways areas should detect unauthorized access and undesired intrusions by means of intelligent and autonomous systems. Specific features of railway environments make many conventional solutions unsuitable. A railway asset—such as a service area, a depot for trains or a railway station—cannot be closed with barriers because they would obstruct the passage of authorized staff or passengers. Obviously, the same rail tracks cannot be physically obstructed by physical barriers, such as sensorized fences or taut wire sensors. Other electronic technologies, such as microwave sensors, electric field sensors or ported coaxial cables, suffer from electromagnetic interferences associated with train transit [125,160].

Recently, the Optoelectronic Division—Engineering Department of the University of Sannio, Benevento proposed an optical fiber security system based on an FBG multiplexing sensing technique to prevent intruders from accessing the railway area without permission. Two technological solutions were developed and tested in realistic operative scenarios. A ribbed mat integrated with FBG strain sensors was installed near the railway track area [161], whilst FBG accelerometers were attached on the rail tracks to recognize intrusion by vibration signature [162]. The results show that the FBG strain sensors in the ribbed mat were able to monitor real-time human activities and the FBG accelerometers were able to monitor a human walking on the rail tracks [161,162]. The proposed technological strategy was adopted to protect specific railways assets, but it is representative of a quite common railway environment for outdoor security applications.

In Figure 21, we illustrate the intrusion detection system installed at the entry point of the EAV railway depot (Ponticelli—Napoli) in more detail. The sensing system was conceived to operate alone to protect a “walkable” entry point or in combination with a thermal/video camera to improve the recognition capability of existing video analysis systems.

The sensing system is composed of a rubber mat with FBG strain sensors integrated on the lower surface of the mat. An intruder walking on the mat represents a load on the upper surface of the soft rubber mat, endowed by a low Young’s modulus. The FBGs, bonded on the ribbed lower surface of the mat, sense the strain associated with the load and convert the strain to a Bragg wavelength shift. The influence area of a single FBG integrated in the mat has been preliminary characterized in the laboratory. Then a network of 75 equally spaced FBGs has been properly deployed under the mat in order to attain a total coverage of the sensorized area without blind zones.

Several trials were carried out in the field to verify the functionality of the system in the presence of intruders walking on the mat. As briefly reported in Figure 22, the experimental results reveal that FBGs-based sensing system enables the detection of an intruder’s break-in. Indeed, the security system is able to detect intrusion events and generate alarms in real time. Furthermore, we show that the system can be empowered by an intelligent processing system to provide further information about the intrusion event. Indeed, by taking into consideration the sensors’ traces, as well as the spatial distribution of the alarms, additional information, such as the presence of multiple intruders and the intruders’ path or velocity can be inferred.

Similarly, for the rail track access protection, we carried out in-field trials at the Maddaloni–Marcianise freight station (Italy) in order to investigate the capability of the sensing system to reveal the presence of an intruder walking close to the rail tracks. Here, the sensing system for the rail track access protection was composed of two FBG accelerometers installed under the tracks [162]. A walking intruder represents a vibration source that generates specific acoustic signatures. Therefore, the FBG accelerometers can detect intruders by means of the acoustic waves generated by footsteps. The experimental results demonstrate that the FBG accelerometer enables the detection of a walking intruder up to 5 m distance from the rail track access.

We definitively demonstrated the capability of optical fiber sensing technology to offer solutions suitable to protect large areas from unauthorized activities in railway restricted-access zones, such as stations, depots or tunnels. Both installed sensing systems based on accelerometers underneath the tracks or FBGs integrated in a rubber mat represent effective security systems able to protect railway areas from unauthorized activities. The experimental results demonstrated that the intrusion detection system is able to detect a human entering in the protected perimeter (across either the rail tracks or the mat). We have demonstrated the capability of the sensing system to successfully accomplish the detection of intrusion events in a real and “open” scenario. Therefore, optical fiber intrusion sensing systems represent a valuable solution for railway security applications.

## 4. Conclusions

This paper provides a comprehensive review of our results of in-field applications of optical fiber sensors for infrastructural monitoring and railways safety. In particular, we have presented innovative systems for weight measurement, wheel defect monitoring and gas detection for the railway industry, based either on point-based or distributed optical fiber sensors. We also presented a system for road traffic monitoring using already installed optical fibers. Finally, we have discussed some innovative systems for anti-intrusion detection in large areas and dangerous gases detection inside tunnels. The list of applications is completed by those reported in refs. [1,2] published in this issue.

The effectiveness of the results above together with other impressive work of the group is proven by the creation of some spin-off companies (OFTEN MEDICA, OPTOSMART, OPTOSENSING, BIOTAG), working in medical applications and in the design and application of point-based and distributed optical fiber sensors, as well as by the construction of the Center of Nanophotonics and Optoelectronics for Human Health and Industrial Applications (CNOS), which will be completed by April 2023. This Center, built by CERICT (the Competence Center on ICT of Regione Campania), will have a surface of about 3.000 square meters. CERICT is a consortium whose members are the seven Universities of Campania (Italy), the National Council of Research of Italy and the Fondazione Pascale, which is one of the highest-regarded Italian Oncological Centers. It is committed to the development of new devices and technologies based on nanophotonics for human health and new industrial applications. A multidisciplinary approach is very often required in order to provide the most suitable solution in a large variety of practical cases. Accordingly, both the team and the technologies are strongly multidisciplinary, as can be seen in Table 1, where we have listed the main technological areas in which the center is divided. One of the CNOS’ central missions is to help new high-tech start-up companies to enter the market, as well.

The main goals of our future activity can be divided along three main lines:To translate the innovative research results into other market products, thus creating new startup companies;To explore new applications to improve both the safety and the security in other fields, such as agrifood, anti-terrorism, biomedical and precision medicine, environment and energy saving;To improve the performance of our devices by increasing the use of both the nanotechnology and the nano materials.

## Figures and Tables

**Figure 1 sensors-23-02558-f001:**
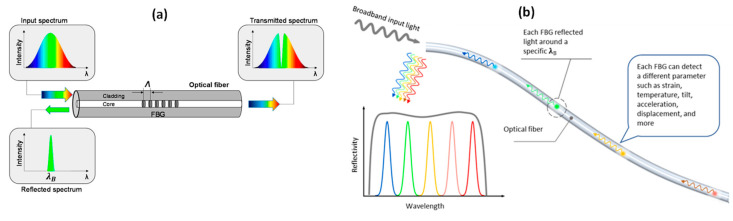
Working principle of fiber Bragg gratings [67,68]: (**a**) schematic of the device and typical reflected and transmitted spectra; and (**b**) schematic of the FBG array in a single fiber by wavelength multiplexing.

**Figure 2 sensors-23-02558-f002:**
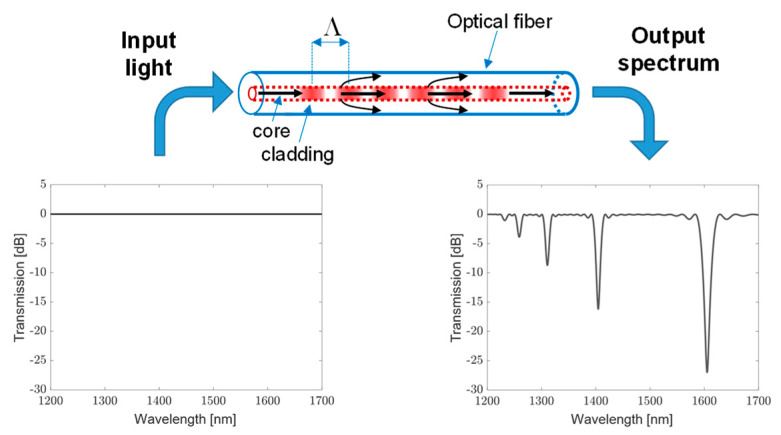
Working principle of long period grating [74,75]: schematic of the device and typical transmission spectrum. Arrows indicate the input and output light of the optical fiber.

**Figure 3 sensors-23-02558-f003:**
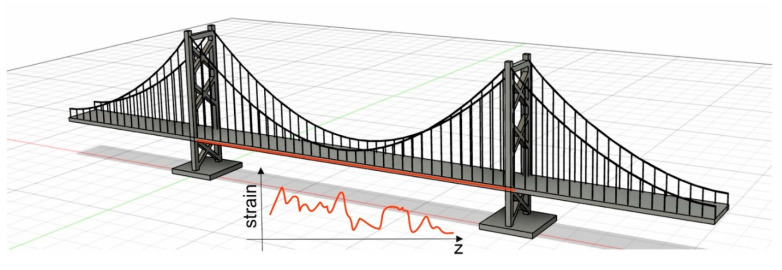
Distributed optical fiber sensors for structural health monitoring.

**Figure 4 sensors-23-02558-f004:**
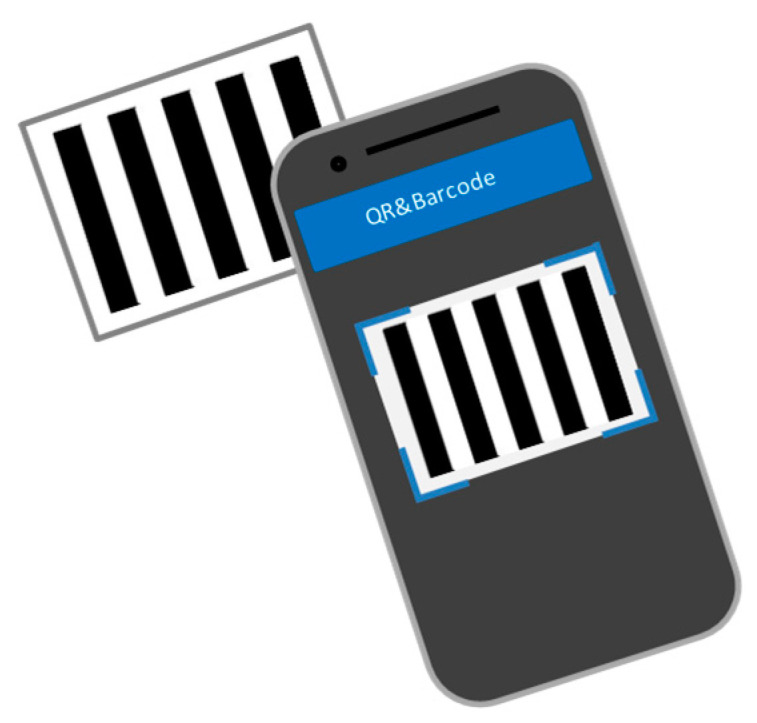
Device schematization.

**Figure 5 sensors-23-02558-f005:**
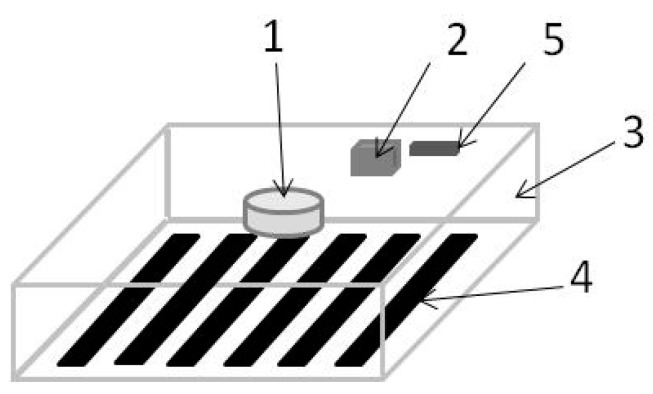
Schematics of the active device: 1. camera, 2. processing unit, 3. cover box, 4. applied image, 5. control unit.

**Figure 6 sensors-23-02558-f006:**
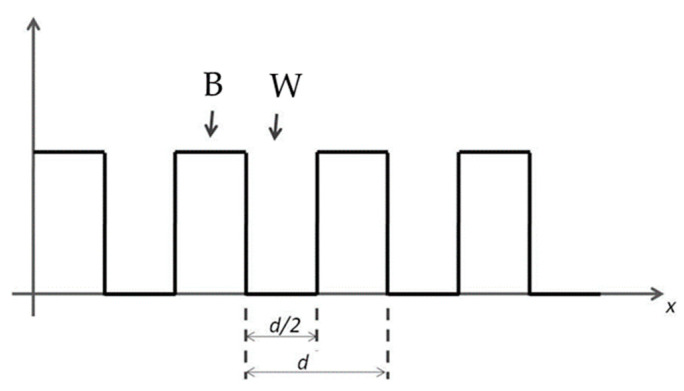
The basic configuration of the barcode.

**Figure 7 sensors-23-02558-f007:**
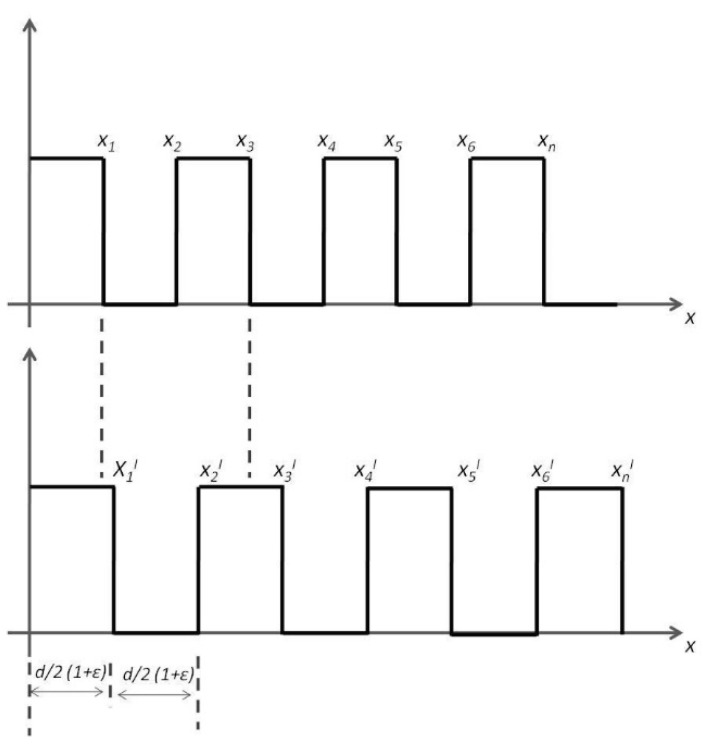
The distribution of white and black lines of the barcode before (on the (**top**)) and after (on the (**bottom**)) the deformation has taken place.

**Figure 8 sensors-23-02558-f008:**
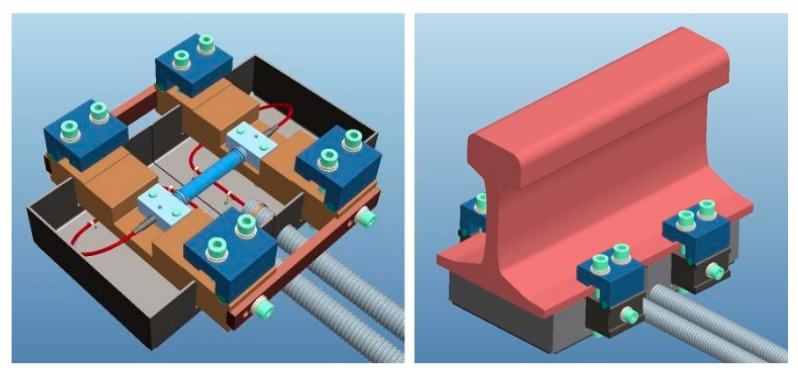
WIM–WILD sensor rendering [133].

**Figure 9 sensors-23-02558-f009:**
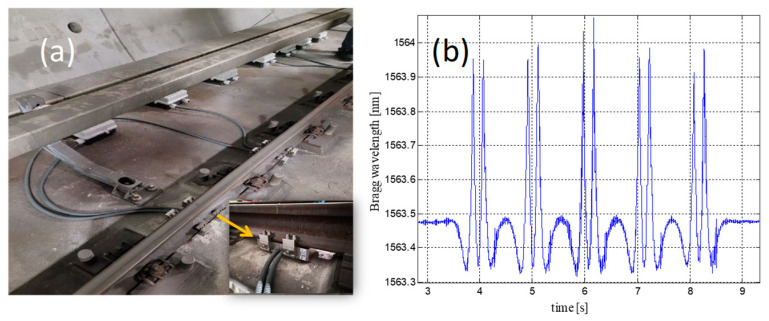
WIM–WILD in-field installation photo (**a**) and typical FBG waveform [133] during a passing train (**b**).

**Figure 10 sensors-23-02558-f010:**
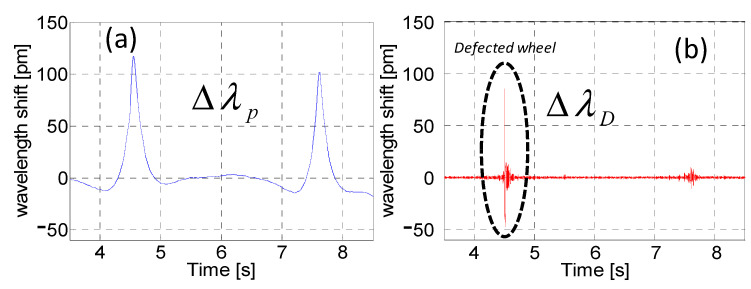
Quasi-static (**a**) and dynamic (**b**) WIM–WILD sensor waveform for a two-axle wagon transit [133].

**Figure 11 sensors-23-02558-f011:**
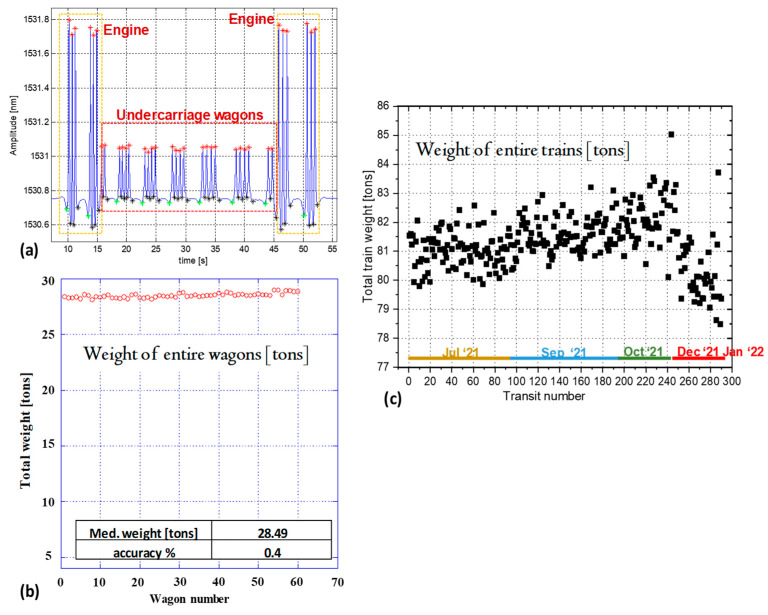
Typical waveform returned by a WIM–WILD sensor (**a**) and WIM results carried out by a system installed in the United Arab Emirates (**b**) and in a Milan subway (**c**). Similar results are reported in [133].

**Figure 12 sensors-23-02558-f012:**
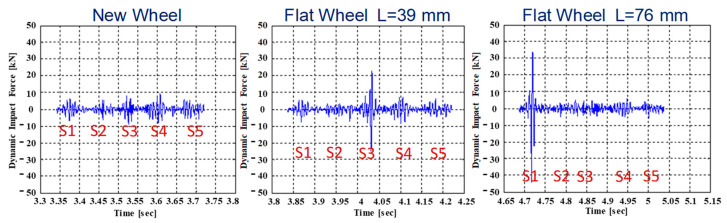
Dynamic impact force as a function of the defect length (for a train speed of 30 km/h). Similar results are reported in [123,127].

**Figure 13 sensors-23-02558-f013:**
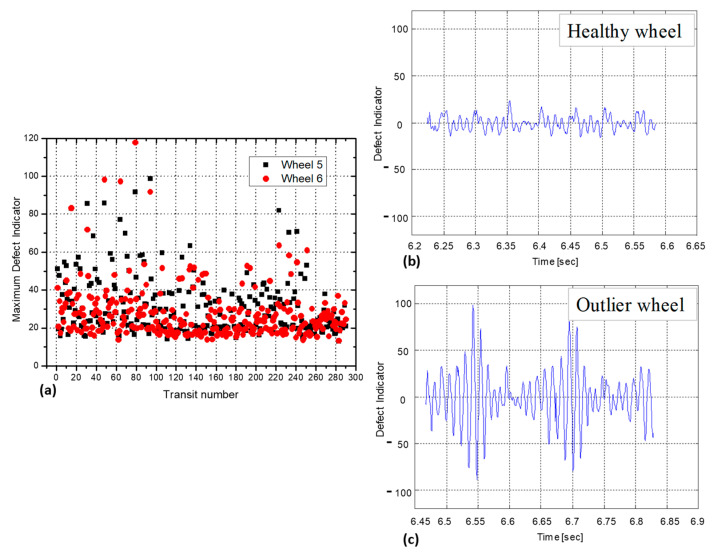
Maximum values of the defect indicator recorded in a Milan subway system (**a**) for 290 passenger trains, typical vibration pattern returned by a healthy (**b**) and outlier (**c**) wheel. Similar results are reported in [133].

**Figure 14 sensors-23-02558-f014:**
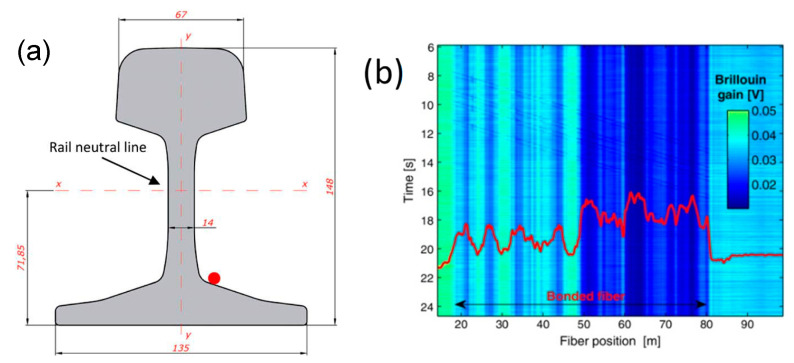
(**a**) Cross-section of the rail, also indicating the position of the glued fiber (red dot), and (**b**) Brillouin gain map as a function of time and position. The superimposed red curve represents the acquired Brillouin frequency shift profile (from Ref. [139]).

**Figure 15 sensors-23-02558-f015:**
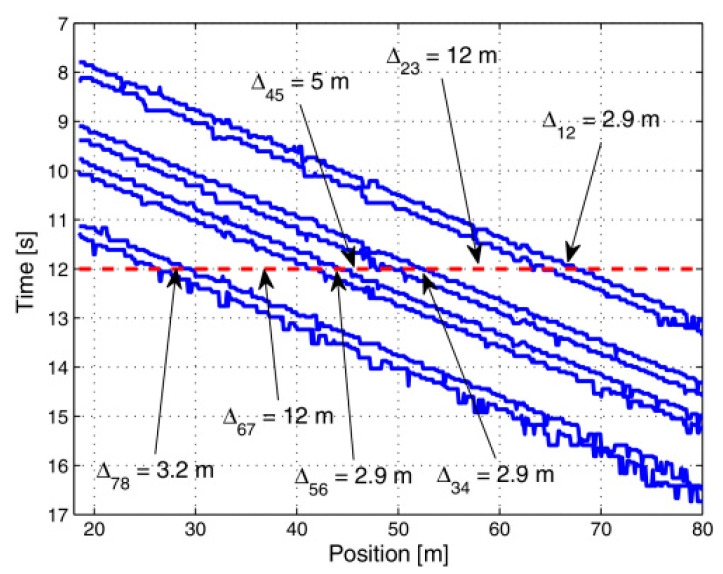
Axle traces retrieved from the acquired data shown in Figure 14b. The estimated axle distances are also reported. The horizontal dashed line indicates the time instant chosen for the axle distance estimation (from Ref. [139]).

**Figure 16 sensors-23-02558-f016:**
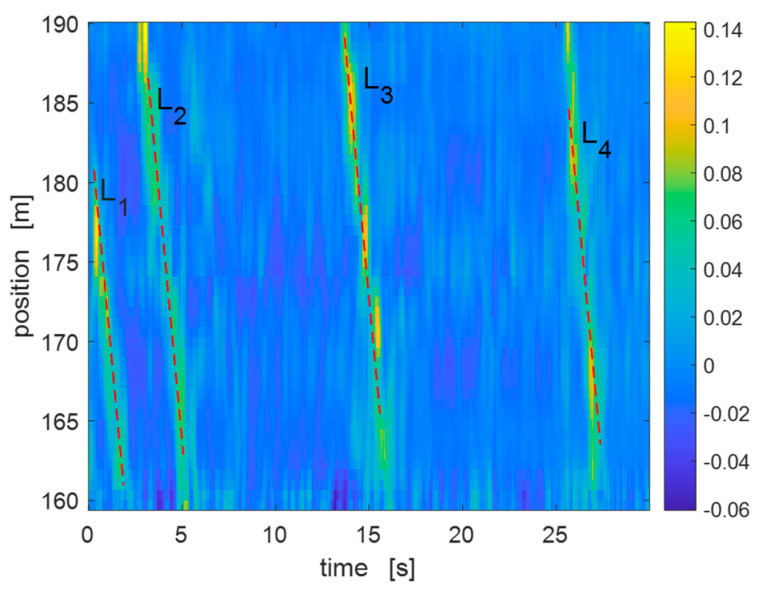
Distributed acoustic sensor recordings superimposed on the lines detected by the Hough transform (from Ref. [144]).

**Figure 17 sensors-23-02558-f017:**
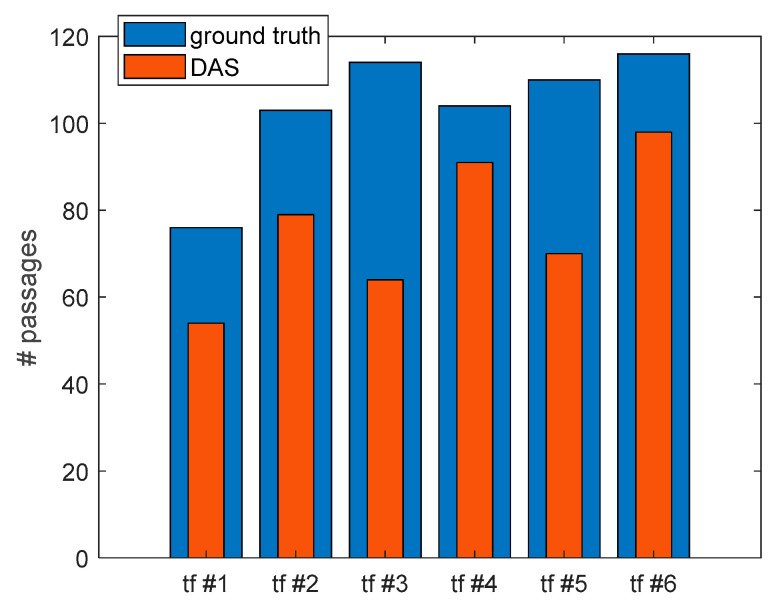
Number of actual (blue bars) and HT-detected (red bars) vehicle passages for each time frame of 10 min (from Ref. [144]).

**Figure 18 sensors-23-02558-f018:**
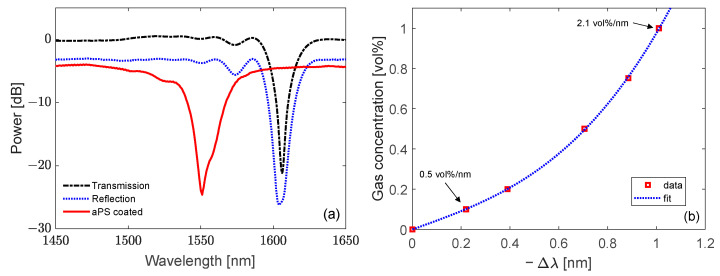
LPG-based gas sensor: (**a**) spectra acquired during fabrication steps (inscription, reflection configuration and with aPS coating); (**b**) response to different butane mixtures below the LEL, in terms of resonance wavelength shift. (Adapted from [147]).

**Figure 19 sensors-23-02558-f019:**
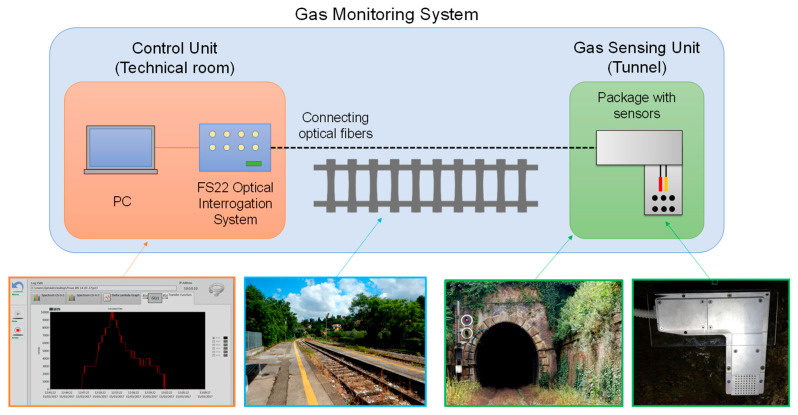
Schematic illustration and pictures of the gas monitoring system for railway tunnel. (Adapted from [147]).

**Figure 20 sensors-23-02558-f020:**
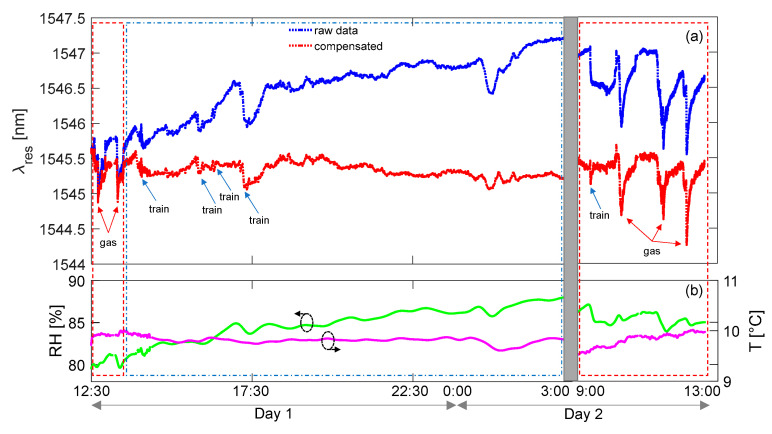
Testing of the liquified monitoring gas detection system: (**a**) response of the sensor in terms of resonance wavelength (raw and compensated data); (**b**) temperature (magenta) and humidity (green) measurements used for the compensation. Red boxes indicate the testing sessions whereas blue boxe is for the line under normal operation (Adapted from [147]).

**Figure 21 sensors-23-02558-f021:**
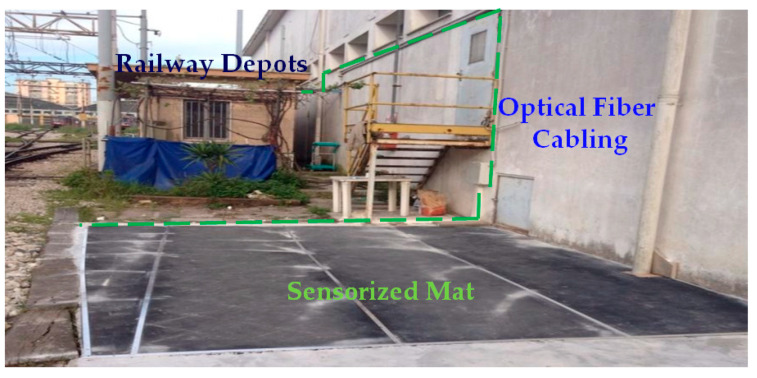
Intrusion detection system in a railway asset installed at the entry point of the Ponticelli depot. Reproduced with permission from ref. [161] (the figure is released under a Copyright Clearance Center’s RightsLink^®^ service).

**Figure 22 sensors-23-02558-f022:**
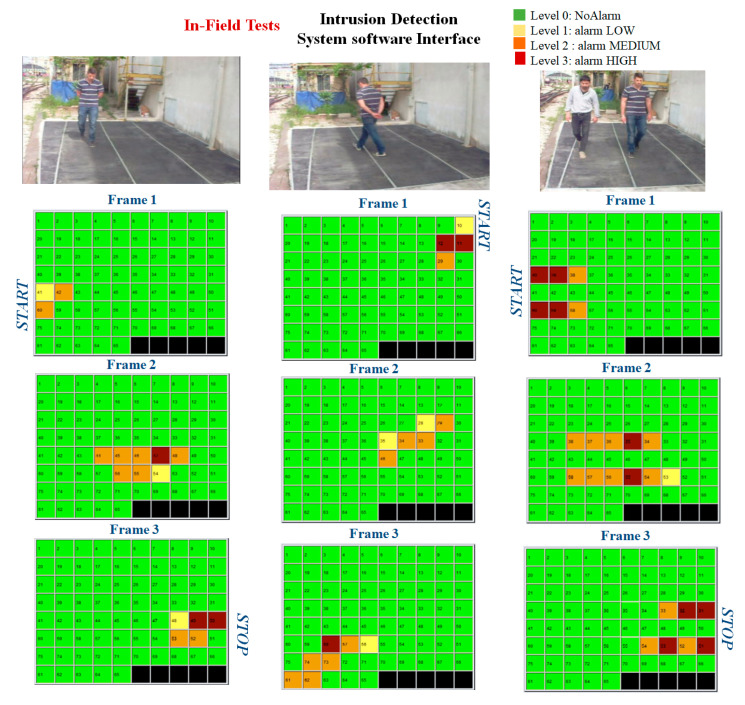
In-field test of the intrusion detection system. Reproduced with permission from ref. [161] (the figure is released under a Copyright Clearance Center’s RightsLink^®^ service).

**Table 1 sensors-23-02558-t001:** Technological areas of the CNOS center.

Nanophotonic laboratory	Chemistry laboratory	Prototyping and smart devices
Biotechnology laboratory	Characterization and test laboratory	Informatic laboratory

## Data Availability

No new data were created or analyzed in this study. Data sharing is not applicable to this article.

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
