# Peer review of "Innovative Photonic Sensors for Safety and Security, Part I: Fundamentals, Infrastructural and Ground Transportations"

_sensors, 2023, doi:10.3390/s23052558_

Round 1

Reviewer 1 Report

The manuscript reviews recent applications of optical fiber sensors for safe and security. In particular, this work focuses on the real time monitoring of fundamentals, infrastructural and ground transportations. After describing the operational principle of these sensors representative and successful applications to various Italian railways are presented, which is consistent with the main topic of this special issue entitled "State-of-the-Art Sensors Technology in Italy 2022". 

The manuscript is well shaped and reads smoothly. The scientific aspects sounds technically. The applications well documented and convincing. 

Overall, I enjoyed reading this paper. As such, I recommend this work for publication. 

Minor comments.

1) Figure 8 (line 433) . "on the right" I guess it should be "left"

2) Sometimes the authors confuse "use" (verb) with "usage" (noun)

3) (Optional) I encourage the authors to expand a bit more the conclusions by adding some prospective.

Author Response

The manuscript reviews recent applications of optical fiber sensors for safe and security. In particular, this work focuses on the real time monitoring of fundamentals, infrastructural and ground transportations. After describing the operational principle of these sensors representative and successful applications to various Italian railways are presented, which is consistent with the main topic of this special issue entitled "State-of-the-Art Sensors Technology in Italy 2022".

The manuscript is well shaped and reads smoothly. The scientific aspects sound technically. The applications well documented and convincing.

Overall, I enjoyed reading this paper. As such, I recommend this work for publication.

We thank the reviewer for his/her positive comment.

1) Figure 8 (line 433). "on the right" I guess it should be "left"

2) Sometimes the authors confuse "use" (verb) with "usage" (noun)

3) (Optional) I encourage the authors to expand a bit more the conclusions by adding some prospective.

We thank the reviewer for his/her comment. The manuscript has been corrected according to his/her suggestions.

Reviewer 2 Report

The review manuscript entitled” Innovative photonic sensors for safety and security. Part I: Fundamentals, Infrastructural and ground transportations”  By  Aldo Minardo et al. reviews the main results of the photonic sensors for safety and security in health care, industrial and environmental applications. The authors have performed quite good reviews on their investigation. However, in the beginning, the article was not well written, but after a while, it became interesting to read and provided enough references and technical details. The review research work is the sum of all photonic sensors for safety and security in health care, industrial and environmental applications. This study is much needed in the fiber optic sensor regime. The language used in this article is reasonably good. I want to address a few queries on this manuscript, which will help improve the quality of the article. Please find the comment below.

1.       On Page no 1, line 34, the Authors state, “In the last three decades” It would be good to provide a reference article on safety and security three decades ago.

2.      On Page no 2, line 45, the author state, “we have realized a large variety of practical applications” What does the author mean by practical applications? 

3.      Page no 2, lines 60-78, the paragraph seems irrelevant here. The paragraph may move to the field result or the end of the manuscript. How is this paragraph related to reviewing the main results of the photonic sensors for safety and security in health care, industrial and environmental applications? 

4.      Page no 2, line 79, Section 2.1, there is not much new information. Why do authors think this section is needed, and would it be good to cite the reference in the figure caption 

5.      Figure 1. (a) Working principle of Fiber Bragg Grating: schematic of the device and typical reflected and transmitted spectra; and (b). [***] . The caption for (b) is missing.

6.      Figure 2. Working principle of Long Period Grating: schematic of the device and typical transmission spectrum. [***]. In the figure labels, the wavelength range (number) is missing. 

7.      On Page 4, line 163, section 2.2, “ In the last decades, distributed optical fiber sensors have emerged as an important tool for the structural health monitoring of bridges, tunnels, aircrafts, etc., as they permit to follow the variations of temperature and strain with a single, conventional optical fiber, 166 with good accuracy and spatial resolution[*]. Please provide a suitable reference if you have any 

8.      On Page 6, line 266, the sensor is wireless and temperature compensated. It can be mounted either embedded (also in buildings of historical and artistic importance) or on sight. In addition, depending on the particular configuration, it can be interfaced with either a PC or a mobile phone.[*]. Please provide a suitable reference if you have any 

9.      Figure 4. Device schematization [*] . Please provide a suitable reference if you have any.

10.  Figure 11. Typical waveform returned by a WIM-WILD sensor (a) and WIM results carried out by 564 a system installed in United Arab Emirates (b) and in Milan subway (c). [*]. Please provide a suitable reference if you have any

11.  On Page 8, line 310, "we refer to a very simple configuration which can be analyzed with a very simple 310 analytical model by which we can easily estimate the main performance"[*]. Please provide a suitable reference if you have any.

12.  Figure 5. Schematics of the active device: 1. Camera, 2. Processing unit, 3. Cover box, 4. Applied image 5. Control unit[*] . Please provide a suitable reference if you have any.

13.  Figure 6. The basic configuration of the BAR code.[*] Please provide a suitable reference if you have any.

14.  On Page 8, line 322, "More specifically, just after the device has been installed on 322 the structure to be controlled, we have [*]. Please provide a suitable reference if you have any.

15.  Figure 7. The distribution of white and black lines of the bar code before (on the top) and after (on 332 the bottom) the deformation has taken place [*]. Please provide a suitable reference if you have any.

16.  On Page no10, line 343 “Equations (8,9) are the characteristic equations of our sensor. [*]. Please provide a suitable reference if you have any.

17.  On Page no10, line 343 "This kind of systems is actually widely used, and they are respectively known as WILD (Wheel Impact Load Measurement) and WIM (Weight in Motion) systems." [*].Please provide a suitable reference if you have any.

18.  Figure 9. WIM-WILD in field installation (left) and typical FBG waveform during a passing train (right). [*]. Please provide a suitable reference if you have any

19.  Figure 12. Dynamic impact force as function of the defect length (for a train speed of 30 km/h). [*]. Please provide a suitable reference if you have any.

Author Response

The review manuscript entitled” Innovative photonic sensors for safety and security. Part I: Fundamentals, Infrastructural and ground transportations” By  Aldo Minardo et al. reviews the main results of the photonic sensors for safety and security in health care, industrial and environmental applications. The authors have performed quite good reviews on their investigation. However, in the beginning, the article was not well written, but after a while, it became interesting to read and provided enough references and technical details. The review research work is the sum of all photonic sensors for safety and security in health care, industrial and environmental applications. This study is much needed in the fiber optic sensor regime. The language used in this article is reasonably good. I want to address a few queries on this manuscript, which will help improve the quality of the article. Please find the comment below.

  1. On Page no 1, line 34, the Authors state, “In the last three decades” It would be good to provide a reference article on safety and security three decades ago.

We thank the reviewer for the comment. A few more references have been added.

  1. On Page no 2, line 45, the author state, “we have realized a large variety of practical applications” What does the author mean by practical applications? 

We thank the reviewer for the comment. The statement has been clarified, replacing the expression “practical applications” with “in-field applications”.

  1. Page no 2, lines 60-78, the paragraph seems irrelevant here. The paragraph may move to the field result or the end of the manuscript. How is this paragraph related to reviewing the main results of the photonic sensors for safety and security in health care, industrial and environmental applications? 

We thank the reviewer for the comment. We agree with the Reviewer and the paragraph has been moved to the end of the manuscript in the conclusion section.

  1. Page no 2, line 79, Section 2.1, there is not much new information. Why do authors think this section is needed, and would it be good to cite the reference in the figure caption.
  2. Figure 1. (a) Working principle of Fiber Bragg Grating: schematic of the device and typical reflected and transmitted spectra; and (b). [***] . The caption for (b) is missing.
  3. Figure 2. Working principle of Long Period Grating: schematic of the device and typical transmission spectrum. [***]. In the figure labels, the wavelength range (number) is missing. 

We thank the reviewer. The section 2 and in particular the section 2.1 briefly presents the theoretical aspects and mechanisms used in different applications, which are discussed in section 3 and in connected Part II and Part III papers. We agree that it does not add new information. However, we find it useful for a less expert reader and for the formulation of a commune language. Finally, the captions of figures 1 and 2 have been corrected and references have been added. Figure 2 has been updated.

  1. On Page 4, line 163, section 2.2, “In the last decades, distributed optical fiber sensors have emerged as an important tool for the structural health monitoring of bridges, tunnels, aircrafts, etc., as they permit to follow the variations of temperature and strain with a single, conventional optical fiber, with good accuracy and spatial resolution[*]. Please provide a suitable reference if you have any 

A proper reference has been added.

  1. On Page 6, line 266, the sensor is wireless and temperature compensated. It can be mounted either embedded (also in buildings of historical and artistic importance) or on sight. In addition, depending on the particular configuration, it can be interfaced with either a PC or a mobile phone.[*]. Please provide a suitable reference if you have any 

A proper reference has been added.

  1. Figure 4. Device schematization [*] . Please provide a suitable reference if you have any.

No reference to past works is available.

  1. Figure 11. Typical waveform returned by a WIM-WILD sensor (a) and WIM results carried out by a system installed in United Arab Emirates (b) and in Milan subway (c). [*]. Please provide a suitable reference if you have any.

No reference to past works is available.

  1. On Page 8, line 310, "we refer to a very simple configuration which can be analyzed with a very simple analytical model by which we can easily estimate the main performance"[*]. Please provide a suitable reference if you have any.

No reference to past works is available.

  1. Figure 5. Schematics of the active device: 1. Camera, 2. Processing unit, 3. Cover box, 4. Applied image 5. Control unit[*] . Please provide a suitable reference if you have any.

No reference to past works is available.

  1. Figure 6. The basic configuration of the BAR code. Please provide a suitable reference if you have any.

No reference to past works is available.

  1. On Page 8, line 322, "More specifically, just after the device has been installed on the structure to be controlled, we have...”. Please provide a suitable reference if you have any.

No reference to past works is available.

  1. Figure 7. The distribution of white and black lines of the bar code before (on the top) and after (on 332 the bottom) the deformation has taken place [*]. Please provide a suitable reference if you have any.

No reference to past works is available.

  1. On Page no10, line 343 “Equations (8,9) are the characteristic equations of our sensor. [*]. Please provide a suitable reference if you have any.

No reference to past works is available.

  1. On Page no10, line 343 "This kind of systems is actually widely used, and they are respectively known as WILD (Wheel Impact Load Measurement) and WIM (Weight in Motion) systems". Please provide a suitable reference if you have any.

A proper reference has been added.

  1. Figure 9. WIM-WILD in field installation (left) and typical FBG waveform during a passing train (right). [*]. Please provide a suitable reference if you have any.

No reference to past works is available.

  1. Figure 12. Dynamic impact force as function of the defect length (for a train speed of 30 km/h). [*]. Please provide a suitable reference if you have any.

No reference to past works is available.

Reviewer 3 Report

This manuscript reviews main results of Innovative photonic sensors in applications for safety and security. This the first part mainly focuses on Fundamentals, Infrastructural and ground transportations, followed by Aerospace and submarine applications (Part II) and Environment Applications (Part III). The authors Aldo Minardo et al. are senior Scientists and engineers from different Universities in Campania, Italy and has been working in this area for the last twenty years. I recommend it for publication in Sensors after the following questions and comments have been addressed satisfactorily.

1.      It would be better to focus on the main results in Part I: Fundamentals, Infrastructural and ground transportations.

Our group, involving researchers from different Universities in Campania, Italy, has been working for the last twenty years in the field of photonic sensors for safety and security in health care, industrial and environment applications. In this manuscript we review the first part of our main results concerning fundamentals and applications for infrastructural, transportation and environment monitoring.

2.      It would be better to modify the manuscript for a better introduction of the research background.

3.      There are two sections namely 3.4. And, is there only one subsection 3.4.1?

4.      Discussion, outlook, and conclusion should be added.

5.      For optical fiber sensors, what is the main bottleneck of this technology?

6.      Can the authors introduce their future research direction or content?

Author Response

This manuscript reviews main results of Innovative photonic sensors in applications for safety and security. This the first part mainly focuses on Fundamentals, Infrastructural and ground transportations, followed by Aerospace and submarine applications (Part II) and Environment Applications (Part III). The authors Aldo Minardo et al. are senior Scientists and engineers from different Universities in Campania, Italy and has been working in this area for the last twenty years. I recommend it for publication in Sensors after the following questions and comments have been addressed satisfactorily.

  1. It would be better to focus on the main results in Part I: Fundamentals, Infrastructural and ground transportations. Our group, involving researchers from different Universities in Campania, Italy, has been working for the last twenty years in the field of photonic sensors for safety and security in health care, industrial and environment applications. In this manuscript we review the first part of our main results concerning fundamentals and applications for infrastructural, transportation and environment monitoring.

We thank the Reviewer. We have slightly modified the abstract of the paper, making it clearer that this is only the first in a series of three companion papers, and that this first one focuses on our main results concerning the infrastructural and transportation monitoring.

  1. It would be better to modify the manuscript for a better introduction of the research background.

In Section 2, we briefly introduce the main concepts of the technologies used for our photonic sensors (fiber Bragg gratings, long period gratings and distributed optical fiber sensors).  A new concept of strain sensors, based on bar and QR codes, is provided as well.

  1. There are two sections namely 3.4. And is there only one subsection 3.4.1?

We thank the Reviewer for letting us note this. We have fixed the section numeration.

  1. Discussion, outlook, and conclusion should be added.

Discussion of the single topics is included in each separated section. A conclusion paragraph has been added, providing details about the main results and future perspective of our research group. 

  1. For optical fiber sensors, what is the main bottleneck of this technology?

We believe that the main bottleneck of fiber-optic sensing technologies is the high cost of the interrogation units. We should also mention that the technology is still somewhat unfamiliar to the user, and hence it requires some basic training before its use. Furthermore, it requires precise installation procedures to get reliable results and special care in fiber optic handling.

  1. Can the authors introduce their future research direction or content?

The main perspectives of our future research direction have been discussed in the conclusion paragraph.

Round 2

Reviewer 2 Report

The authors have submitted a revised review manuscript entitled” Innovative photonic sensors for safety and security. Part I: Fundamentals, Infrastructural and ground transportations by Aldo Minardo et al. reviews the main results of the photonic sensors for safety and security in health care, industrial and environmental applications. The authors have significantly improved the review manuscript by illustrating their research work in terms of modifications, figures and contents. Also, the authors have given satisfactory responses to the comment raised. However, it would be nice to have some references, especially in the equations and the graph (figure 13). Since it is a review manuscript, the equations and results must have been published already. The article will be the appropriate form for publication after considering these queries.

Author Response

We thank the Reviewer for his/her positive comments. In the revised paper, we have added new references, especially for the equations and figures. 

Reviewer 3 Report

I recommend it for publication in Sensors now.

Author Response

We thank the Reviewer for his/her positive comments.